# The Cellular Prion Protein and the Hallmarks of Cancer

**DOI:** 10.3390/cancers13195032

**Published:** 2021-10-08

**Authors:** Sophie Mouillet-Richard, Alexandre Ghazi, Pierre Laurent-Puig

**Affiliations:** 1Centre de Recherche des Cordeliers, Université de Paris, INSERM, Sorbonne Université, F-75006 Paris, France; alexandre.ghazi@parisdescartes.fr (A.G.); pierre.laurent-puig@parisdescartes.fr (P.L.-P.); 2Department of Biology, Institut du Cancer Paris CARPEM, APHP, Hôpital Européen Georges Pompidou, F-75015 Paris, France

**Keywords:** cellular prion protein, cancer hallmarks, signalling pathways

## Abstract

**Simple Summary:**

The cellular prion protein PrP^C^ is best known for its involvement, under its pathogenic isoform, in a group of neurodegenerative diseases. Notwithstanding, an emerging role for PrP^C^ in various cancer-associated processes has attracted increasing attention over recent years. PrP^C^ is overexpressed in diverse types of solid cancers and has been incriminated in various aspects of cancer biology, most notably proliferation, migration, invasion and metastasis, as well as resistance to cytotoxic agents. This article aims to provide a comprehensive overview of the current knowledge of PrP^C^ with respect to the hallmarks of cancer, a reference framework encompassing the major characteristics of cancer cells.

**Abstract:**

Beyond its causal involvement in a group of neurodegenerative diseases known as Transmissible Spongiform Encephalopathies, the cellular prion protein PrP^C^ is now taking centre stage as an important contributor to cancer progression in various types of solid tumours. The prion cancer research field has progressively expanded in the last few years and has yielded consistent evidence for an involvement of PrP^C^ in cancer cell proliferation, migration and invasion, therapeutic resistance and cancer stem cell properties. Most recent data have uncovered new facets of the biology of PrP^C^ in cancer, ranging from its control on enzymes involved in immune tolerance to its radio-protective activity, by way of promoting angiogenesis. In the present review, we aim to summarise the body of literature dedicated to the study of PrP^C^ in relation to cancer from the perspective of the hallmarks of cancer, the reference framework defined by Hanahan and Weinberg.

## 1. Introduction

The cellular prion protein PrP^C^ was discovered in the mid-1980s as the normal counterpart of the scrapie prion protein, denoted PrP^Sc^, itself responsible for the development of a group of fatal neurodegenerative diseases known as Transmissible Spongiform Encephalopathies or prion diseases [1]. PrP^C^, encoded by the *PRNP* gene located on chromosome 20 in humans, is a ubiquitous protein that is highly conserved from fish to mammals [2]. It is a small glycoprotein of 253 amino acids subject to various types of post-translational modifications: removal of a N-terminal signal peptide responsible for the trafficking of the protein to the endoplasmic reticulum for subsequent maturation, replacement of the C-terminus with a glycosyl-phosphatidylinositol (GPI) moiety that allows PrP^C^ anchoring at the extracellular plasma membrane, formation of a disulfide bridge and potential N-glycosylation on two Asparagine residues (reviewed in [3]). Beyond being majorly GPI-anchored at the cell membrane, PrP^C^ may additionally exist as two topological transmembrane variants with either the N-terminus (CtmPrP) or C-terminus (NtmPrP) portion in the cytosol, possibly accounting for the interaction with cytosolic partners [3]. From a structure–function point of view, PrP^C^ is composed of a N-terminal, intrinsically disordered domain, also referred to as “flexible tail”, a central hydrophobic domain and a C-terminal globular domain (Figure 1). A notable feature within the N-terminal domain is the presence of four histidine-containing octapeptide tandem repeats, which are involved in the binding of divalent ions such as copper or zinc, themselves promoting the endocytosis of PrP^C^ [4]. In addition to full-length isoforms, several proteolytic processes can generate various truncated or soluble forms of PrP^C^ (reviewed in [5]). First, the so-called alpha-cleavage, occurring at position 111/112, generates a N-terminal fragment termed N1 and a C-terminal, GPI-anchored fragment termed C1 (Figure 1). This alpha-cleavage occurs under physiological conditions and influences the endocytosis of PrP^C^ as well as its interaction with diverse partners [5]. The beta-cleavage generating N2 and C2 fragments takes place within the octarepeat region (Figure 1). It is mostly described as a reactive oxygen species-dependent reaction sustaining the protective role of PrP^C^ against oxidative stress [5]. Finally, a far C-terminal cleavage is responsible for the production of “shed PrP^C^”, which nearly encompasses the full sequence of PrP^C^ (Figure 1). Shed PrP^C^ is, however, distinct from soluble PrP^C^ as it results from the phospholipase-C mediated hydrolysis of the GPI anchor [5]. Altogether, miscellaneous glycosylation and proteolytic processes in fine generate a variety of PrP^C^ isoforms that may underlie its wide range of functions [6]. One should further bear in mind that the various soluble isoforms, N1, N2, shed PrP^C^ as well as phospholipase-C-released PrP^C^, have the capacity to signal to neighbouring cells (see [5] for review). This also holds true for exosomal PrP^C^, as will be discussed below. 

The variety of PrP^C^ isoforms may explain why PrP^C^ has been ascribed a plethora of functions, ranging from broad roles in the physiology of the central nervous system, resistance to various types of stresses, cell fate and differentiation, cell adhesion and cell signalling [6]. Unravelling the physiological roles exerted by PrP^C^ has long emerged as a powerful strategy to understand how the corruption of these functions may contribute to pathological contexts, not only prion diseases [7] but also other disorders, including Alzheimer’s disease, immune disorders or cancer [8]. Obviously, the research dedicated to prion and cancer has lagged behind that of neurodegeneration, but this field is now becoming the focus of growing interest. We here chose to provide a comprehensive review of the current knowledge relating to PrP^C^ in cancer through the lens of the hallmarks of cancer. As such, the contribution of PrP^C^ to the emergence and/or maintenance of cancer stem cell properties or the potential therapeutic strategies to target this protein in cancer will not be discussed here and have been covered by several reviews [9,10,11,12]. The hallmarks of cancer are a reference framework introduced by Hanahan and Weinberg over 20 years ago [13] and further refined in 2011 [14]. It summarises the fundamental capacities endowing cancer cells with the ability to develop and escape control by the organism. In this review, we will summarise the overall data relating to the biology of PrP^C^ in cancer according to each hallmark, except the enabling replicative immortality hallmark due to a lack of data on this axis. We further highlight some findings pertaining to the physiological function of PrP^C^ and discuss their potential implications in the field of cancer. 

## 2. Sustaining Proliferative Signalling

Since sustained proliferative capacity arguably represents one of the most fundamental traits of cancer cells, the contribution of PrP^C^ to cancer cell proliferation has been extensively studied. The first demonstration that PrP^C^ drives the proliferation of cancer cells was brought by the team of Daiming Fan using the SGC7901 and AGS gastric cancer cell lines [15]. In the low PrP^C^-expressing SGC7901 cells, PrP overexpression promoted an increase in cell proliferation in vitro, as well as tumour growth in xenografted nude mice [15]. To the opposite, PrP^C^ silencing in the high PrP^C^-expressing AGS cell line triggered a reduction in their proliferative index [15]. At a mechanistic level, PrP^C^ was shown to foster the transition from the G0/G1 to the S phase and to transcriptionally control the expression of Cyclin D1 via the PI3/AKT signalling pathway. From a structure-function point of view, it is interesting to note that the action of PrP^C^ requires the presence of the N-terminal domain of the protein [15], in line with the importance of this region in the coupling of PrP^C^ to PI3K/AKT signalling in non-tumoral cells [16]. 

Similarly, Li and colleagues documented a reduction in cell proliferation and in vivo tumour growth upon PrP^C^ silencing in the pancreatic cancer cell lines BxPC3 and Pan 02.03 [17]. In a follow-up study based on the same cellular models together with the Capan-1 pancreatic cell line, PrP^C^ was found to control the levels of Ki67 [18], a key marker of cell proliferation [19]. Interestingly, the decrease in cell proliferation observed after PrP^C^ silencing in Capan-1 cells was abrogated upon a concomitant overexpression of the activated form of NOTCH1 [18]. This observation has to be brought together with the PrP^C^-dependent control on the Notch pathway that we documented in neural stem and progenitor cells [20]. A correlation between PrP^C^ and Ki67 levels was also reported by Lopes et al. in a large cohort of patients with glioblastoma [21]. As with gastric and pancreatic cell lines, the latter study demonstrated a pro-proliferative action of PrP^C^ in the U87 glioma cell line and corresponding xenografts, which was dependent upon the interaction of PrP^C^ with its ligand STI1 [21]. The proliferative role of PrP^C^ in glioblastoma was also confirmed in U87 cells grown as spheres to mimic glioblastoma stem cells [22] as well as in primary tumour cells [9]. In addition, PrP^C^ expression was reported to vary according to the cell cycle in U87 glioma cells with significantly higher levels in the G2/M versus G1/S phase [23]. The PrP^C^-dependent control of proliferation was also exemplified in schwannoma [24] and colorectal cancer [25,26,27,28,29,30]. In the context of colorectal cancer, we notably brought to light a PrP^C^-dependent activation of the integrin linked kinase (ILK) that relays its control on cell proliferation [26]. Adding another layer of complexity to the picture, Yun et al. recently reported that the proliferation of various colorectal cancer cells can be sustained by exosomes derived from the same cells grown under hypoxia in a PrP^C^-dependent manner [31]. In this setting, two non-mutually exclusive mechanisms may be at play: the proliferation of recipient cells may be directly regulated by exosomal PrP^C^, the level of which is increased following hypoxia [31], or it may additionally depend upon other exosomal proteins whose abundance in exosomes is influenced by the expression of PrP^C^ in cancer cells. In this respect, it is also worth noting that PrP^C^ regulates the balance between exosome biogenesis and autophagy [32]. Thus, we may surmise that high PrP^C^-expressing cancer cells produce abundant levels of exosomes, enriched in PrP^C^, that may sustain their proliferation in an autocrine and paracrine manner, especially in a hypoxic environment. 

We many finally note that the PrP^C^-dependent regulation of proliferation in the context of cancer may be viewed as a gain of its normal physiological function resulting from its overexpression. Indeed, the contribution of PrP^C^ to normal cell proliferation has been extensively documented (reviewed in [6]), most notably in the context of stem cells (reviewed in [11]). The physiological PrP^C^-dependent regulation of proliferation may involve a modulation of the Epidermal Growth Factor Receptor (EGFR) activity as described by Llorens et al. [33]. Incidentally, PrP^C^ and EGFR were shown to co-localise and to interact, as inferred by co-immunoprecipitation experiments, in the HT29 colorectal cell line [34]. These overall observations warrant investigating the signalling pathways through which PrP^C^ sustains the proliferation of cancer cells and its potential functional interactions with growth factor receptors. 

## 3. Evading Growth Suppressors

This hallmark corresponds to the ability of cancer cells to circumvent anti-growth signals [14]. They may do so by bypassing the activity of suppressors of proliferation such as TP53 and RB, encoding p53 and the retinoblastoma-associated protein, respectively, evading mechanisms of contact inhibition and/or corrupting anti-growth signalling circuitries such as the Transforming Growth Factor β (TGFβ) pathway. Although a direct contribution of PrP^C^ to growth suppressors evasion has yet to be fully investigated, some observations are worth considering. 

First, several studies have uncovered a modulation of p53 expression and activity by PrP^C^ and its proteolytic fragments (reviewed in [35]). On the one hand, the full length PrP^C^ was shown to up-regulate p53 activity and mRNA levels in neuronal cells upon exposure to the apoptotic inducer staurosporine, thereby sensitising cells to cell death [36]. This also holds true for the cleaved C1 fragment of PrP^C^, as shown in HEK293 cells [37]. On the other hand, the N1 soluble fragment of PrP^C^ was found to exert an opposite effect on p53 and to protect cells from the full length and C1 PrP^C^-mediated potentiation of cell death [38]. Importantly, this set of studies, as well as others [39,40], indicated that overexpressed PrP^C^ has no impact on the p53 pathway in basal conditions, i.e., in the absence of pro-apoptotic signals.

Nevertheless, an activation of p53 signalling was reported upon PrP overexpression in skeletal muscle [41]. In a more cancer-relevant context, Liang et al. documented that PrP^C^ silencing promotes an increase in the expression levels of p53 in the gastric cancer cell line AGS, while an opposite effect was obtained upon PrP^C^ overexpression [42]. Thus, until now, these scarce studies have provided only a glimpse of the potential regulation of p53 by PrP^C^, which may notably depend upon the relative abundance of its different isoforms.

Regarding contact inhibition, an interesting observation is the upregulation of PrP^C^ in various types of Merlin-deficient tumours, including schwannoma and mesothelioma [24]. Merlin, encoded by the neurofibromatosis type 2 (NF2) gene, is a well-described regulator of cell–cell attachment, whose loss of function allows cells to evade contact inhibition [43]. Of note, we recently demonstrated that PrP^C^ levels positively control the phosphorylation of NF2 on serine 518, itself negatively regulating NF2 activity, in colorectal cancer [26]. Mechanistically, PrP^C^ operates via ILK [26], previously described as an upstream regulator of the NF2-Hippo pathway in various types of cancer cells [44]. Accordingly, we documented that the PrP^C^-ILK module promotes the activation of YAP/TAZ [26], the two transcriptional effectors of the Hippo pathway, which play a key role in promoting the poor-prognosis mesenchymal subtype of colorectal cancer [28,45]. Since YAP/TAZ are major orchestrators of organ growth and contact inhibition [46], their upstream regulation by PrP^C^ clearly delineates a link between PrP^C^ and contact inhibition. Furthermore, because NF2 controls the cell surface availability of various growth factor receptors [43], it will be interesting for future studies to evaluate the impact of its negative regulation by PrP^C^ on growth factor receptor signalling. 

Finally, regarding TGFβ, we recently demonstrated that PrP^C^ controls the soluble levels of TGFβ in the supernatant of colorectal cancer cells [28]. Conversely, PrP^C^ levels are increased in response to TGFβ [28]. Although the mechanisms involved still require further investigation, we were able to show that the PrP^C^-TGFβ axis contributes to the expression of several markers that specify the mesenchymal subtype of colorectal cancer, including that of ZEB1, a master regulator of Epithelial to Mesenchymal Transition (EMT) [28]. These observations call for a better understanding of the interplay between PrP^C^ and TGFβ, considering the major role played by TGFβ not only in the poor prognosis subgroup of colorectal cancer [47] but more widely in various aspects of high-grade malignancy across cancer [48]. 

## 4. Resisting Cell Death

Resistance to apoptosis was the first hallmark to be connected with PrP^C^ nearly 20 years ago, as *PRNP* transcripts were found to be upregulated in adriamycin-resistant SGC7901 gastric cancer cells as compared to the parental cell line [49]. That elevated PrP^C^ may confer resistance to anticancer agents was soon confirmed by the demonstration of a causal relationship between increased PrP^C^ expression and resistance to tumour necrosis factor-α (TNFα) in the MCF7 breast cancer cell line [50]. The involvement of PrP^C^ in the resistance of cancer cells to cell death-inducing signals has been extensively studied by employing diverse experimental paradigms. The first set of data provided evidence for an upregulation of PrP^C^ expression in drug-resistant contexts, as in a seminal Zhao study [49]. For instance, *PRNP* gene expression [51] and PrP^C^ protein levels [52] were found to be upregulated in adriamycin-resistant MCF7 breast cancer cells, as well as in SNU-5C colorectal cancer cells resistant to 5-fluorouracil (5-FU) or oxaliplatin [53]. Zhuang et al. further showed a dose-dependent increase in PrP^C^ expression in U87 and U251 glioblastoma cells in response to temolozomide [23]. Moreover, we reported that high expression levels of PrP^C^ are associated with 5-FU resistance in a panel of colorectal cancer cell lines [28]. On the other hand, PrP^C^ overexpression was reported to induce resistance to adriamycin, vincristine, etoposide, 5-FU and cisplatin in SGC7901 gastric cancer cells [54] and resistance to TNFα [30] or adriamycin [25] in LS174 colorectal cancer cells. Conversely, silencing of PrP^C^ increased the sensitivity of MKN28 gastric cancer cells to adriamycin, vincristin, etoposide, 5-FU and cisplatin [55] and that of 5-FU-resistant SNU-5C/FUR [56] or MDST8 [28] colorectal cancer cells to 5-FU. It also cancelled the protection of hypoxia against TRAIL-mediated cell death in HCT116 colorectal cancer cells, while PrP^C^ overexpression conferred resistance to TRAIL-induced cytotoxicity in normoxic conditions [57]. These observations are summarised in Figure 2.

From a translational perspective, it is interesting to note that the relationship between PrP^C^ levels and chemoresistance was confirmed in patients. Indeed, an increased PrP^C^ expression was reported in recurrent versus primary lesions of patients with glioblastoma, following combined temolozomide and radiation therapy [23]. In the same line, Yun et al. reported that plasma levels of PrP^C^ are higher in colorectal cancer patients having received chemotherapy versus untreated patients [31]. Moreover, high PrP^C^ levels in gastric cancer patients were found to be associated with a poor response to chemotherapy [58].

At a molecular level, PrP^C^ may promote chemoresistance through the upregulation of MDR1, which encodes the P-glycoprotein, a transporter responsible for the efflux of anti-cancer drugs, as shown in SGC7901 gastric cancer cells [54]. The PrP^C^-dependent control on MDR1 appears to be mediated by the PI3K-Akt pathway [59] and to necessitate the octarepeat-rich N-terminal domain of PrP^C^ [60], although the deletion of a single octarepeat appears without effect [61]. The recruitment of the PI3K-Akt cascade downstream from PrP^C^ may rely upon its interaction with the 37kD laminin receptor precursor protein (37LRP), as suggested by Luo et al. [62]. A different, non-mutually exclusive mechanism is the positive regulation of the anti-apoptotic effector Bcl2, which may not only occur in gastric [42,54,58], breast [51] and glioma [63] cancer cells but is also a well-described pathway relaying the cell-survival physiological activity of PrP^C^ (reviewed in [64]) and was even recently documented in the context of liver metabolism [65]. Other effects of PrP^C^ on apoptotic effectors include the upregulation of survivin, cIAP-1 and XIAP levels in colorectal cancer cells [25] or the sequestration of the pro-apoptotic factor Par4 in glioblastoma cells [23]. Finally, Wiegmans et al. described a novel mechanism whereby soluble PrP^C^ promotes the resistance of breast cancer cells to adriamycin through the direct binding and sequestration of the drug [66].

Beyond conferring resistance to anticancer drugs, PrP^C^ was recently reported to protect breast and colorectal cancer cells from irradiation-induced toxicity [67]. Accordingly, *PRNP* expression levels were found to be increased following radiation treatment of breast or rectal tumours [67]. As with other PrP^C^-related processes, the radioprotective function of PrP^C^ is not restricted to cancer cells but was also exemplified in hematopoietic progenitor cells [68]. PrP^C^ actually appears to confer a broad resistance to genotoxic stress, in part by potentiating the activity of the APE1 endonuclease, a major player in DNA repair [69].

Overall, the relationship between PrP^C^ and resistance to cell death in cancer is multifaceted and can be viewed as an exacerbation of one of its diverse physiological functions. 

## 5. Inducing Angiogenesis

The link between PrP^C^ and angiogenesis emerged almost two decades ago with several studies showing that PrP^C^ is expressed and released by endothelial cells and that its levels are increased after ischemic injury (reviewed in [70]). These observations were subsequently refined with the demonstration that PrP^C^-deficient mice submitted to cerebral ischemia exhibit poorer recovery as compared with wild-type mice, including reduced neo-angiogenesis [71]. In that study, the authors suggested that PrP^C^ operates, at least in part, by preventing the degradation of HIF1α by the proteasome (see Figure 3) [71]. 

In line with this, Alfaidy et al. suggested that PrP^C^ is involved in placental angiogenesis by controlling the proliferation, migration and tube-like organisation of trophoblastic cells [72]. This study further showed an upregulation of PrP^C^ in response to hypoxia [72], which is now quite well established under various paradigms (reviewed in [73]). The HIF1α-dependent control on PrP^C^ expression actually extends to the stabilisation of the protein through the sequestration of the E3 ligase GP78 by hypoxia-induced HSPA1L (Figure 3), as shown by Lee et al. in colorectal cancer cells [56]. Very recently, these data were expanded with the demonstration that the levels of PrP^C^ released in exosomes of colorectal cancer cells are increased under hypoxia [31]. In accordance with the broad role of hypoxia in tumour-associated angiogenesis [74] and the contribution of exosomes to this process [75], Yun et al. found that exosomes derived from colorectal cancer cells grown under hypoxia promote the proliferation, migration, invasion and permeability of human umbilical vein endothelial cells (HUVECs) (Figure 3) [31]. This seminal study provides the first direct evidence for a contribution of PrP^C^ to cancer-associated angiogenesis. Other indirect support is brought by the downstream effectors of PrP^C^ in cancer cells. For instance, we showed that, in colorectal cancer, PrP^C^ controls the expression of Platelet-Derived Growth Factor C (PDGFC) (Figure 3) [28], a stimulator of angiogenesis [76]. It also activates YAP and TAZ [28], the two main effectors of the Hippo pathway, which have been shown to promote vascular mimicry, a process whereby cancer cells themselves, instead of endothelial cells, form angiogenic tubules to supply blood to the tumour [77]. 

Finally, we may note that the PrP^C^ paralogue Doppel has been incriminated in both developmental [78] and cancer-associated angiogenesis [79]. As Doppel resembles the C-terminal moiety of PrP^C^ [80], whether PrP^C^ or its cleaved fragments recapitulate the functional interaction with VEGFR2 reported for Doppel in tumour endothelial cells [79] seems worth investigating.

Altogether, the contribution of PrP^C^ to cancer-associated angiogenesis is only beginning to be unveiled and will certainly be an important axis for future research.

## 6. Activating Invasion and Metastasis 

A most notable hallmark of cancer cells is their ability to disseminate to distant organs. One major biological process sustaining invasion and metastasis is a transcriptional program referred to as EMT, whereby cells lose cell–cell contacts and acquire the capacity to migrate and degrade the surrounding matrix for dissemination [81]. The EMT program is orchestrated by major transcription factors, such as SNAIL, SLUG, TWIST, ZEB1 or ZEB2 [81]. There are multiple lines of evidence indicating that PrP^C^ promotes the invasion and migration of cancer cells and controls some EMT-associated features. Indeed, silencing of PrP^C^ in adryamicin-resistant MCF7 breast cancer cells reduces their invasion and migration, as well as the expression of two metalloproteases MMP2 and MMP9 [52]. Conversely, PrP^C^ overexpression in MCF7 cells enhances their invasion and migration as well as the expression and activity of MMP9 [82]. Similarly, depleting PrP^C^ in U87 glioma cells reduces their migration on laminin [22]. In the Capan-1 pancreatic cell line, Wang et al. found that the reduction in cell migration induced by the depletion of PrP^C^ was rescued upon overexpression of the activated form of NOTCH1 [18]. In addition, PrP^C^-silencing in SGC7901 or MKN45 gastric cancer cells caused a reduction in their invasion, their expression of MMP11, as well as their ability to metastasise to the liver after tail vein injection [83]. In the context of lung cancer, higher PrP^C^ levels were measured in a panel of invasive versus non-invasive lung cancer cell lines [84]. Furthermore, PrP^C^ was shown to control the invasive and migratory properties of CL1-5 cells via a JNK pathway and the knock-down of PrP^C^ reduced their metastatic potential in vivo [84]. Likewise, various studies based on gain and loss of function experiments in colorectal cancer cell lines confirmed the pro-invasive and pro-migratory action of PrP^C^ [25,26,85]. In line with this, Go et al. depicted an increased invasion and migration in the PrP^C^-positive versus the PrP^C^-negative fraction of 5-FU resistant SNU-5C/FUR cells [27]. Finally, the presence of PrP^C^ together with that of the cancer stem cell marker CD44 at the cell surface of primary colorectal cancer cells was found to control their migration in vitro as well as their metastatic potential after injection in the cecal wall [86]. Several studies have further documented a link between PrP^C^ and EMT. Thus, in primary colorectal cancer cells, PrP^C^ was found to control the expression of the EMT transcription factor TWIST and that of N-cadherin while repressing that of E-cadherin [86]. Similar findings were obtained with HT29 colorectal cancer cells grown under hypoxia [56]. Likewise, we documented that PrP^C^ controls the expression of the EMT transcription factor ZEB1 in colorectal cancer cell lines and that *PRNP* gene expression is significantly correlated with an EMT signature in both colorectal cancer patients and cell panels [28]. These overall findings are summarised in Figure 4 (top panel).

At a mechanistic level, several findings are worth noting. Lacerda et al. found that the PrP^C^ pro-invasive action in colorectal cancer cells depends upon its interaction with its ligand STI1 [87]. Moreover, we recently highlighted the importance of the PrP^C^-ILK coupling in the invasive and migratory properties of the MDST8 colorectal cancer cell line [26]. On the other hand, in melanoma cells, where PrP^C^ is mainly found as a pro-PrP isoform retaining its C-terminus instead of a GPI anchor, invasion and migration depends upon the interaction of pro-PrP with Filamin-A [88], which was shown to form a complex with pro-PrP and NOTCH1 in pancreatic cancer cells [18]. Yun et al. further demonstrated that exosomes derived from various colorectal cancer cells grown under hypoxia can sustain their own invasion and migration in a PrP^C^-dependent manner (see Figure 4, bottom panel) [31].

From a clinical point of view, several studies have depicted a correlation between high levels of tumour PrP^C^—as inferred through immuno-histochemistry—and metastasis in patients with breast [89], gastric [83,90] and colorectal [31,56,86] cancer. In the same line, Lin et al. depicted increased levels of tumour PrP^C^ in patients with invasive versus in situ lung adenocarcinoma [84]. Moreover, in colorectal cancer, we reported an enrichment in the expression of the *PRNP* gene in the mesenchymal subtype [28], itself associated with increased progression to advanced stages [91]. We further documented that plasma levels of PrP^C^ have prognostic value in terms of disease control in metastatic colorectal cancer patients [28]. Thus, PrP^C^ is unambiguously associated with this hallmark of cancer.

## 7. Reprogramming of Energy Metabolism (Emerging Hallmark) 

Metabolic reprogramming features as one of the emerging hallmarks of cancer [14], which has taken centre stage over the past decade [92]. It is clear that studies addressing the question as to whether PrP^C^ may influence the metabolism of cancer cells are very scarce. Most data come from the work by Li et al. who found a regulation of the expression of GLUT1, encoding the glucose transporter 1, downstream from PrP^C^ in the DLD-1 colorectal cancer cell line [29]. Accordingly, the authors demonstrated that PrP^C^ depletion reduces glucose uptake and the glycolytic rate of colorectal cancer cells [29]. From a translational point of view, it is worth noting that the expression levels of PrP^C^ and GLUT1 were correlated in colorectal cancer patients [29]. These findings actually recall the identification of both PrP^C^ and GLUT1 as specific cell-surface biomarkers of the adenoma-to-carcinoma transition in colorectal cancer [93]. Aside from cancer, the link between PrP^C^ and glucose uptake is further strengthened by the work of Ashok et al. based upon a comparison of various tissues of mice deficient in PrP^C^ versus their wild-type counterparts for the expression of glucose transporters [94]. However, in apparent contradiction with the data obtained in cancer cells, the absence of PrP^C^ was associated with increased GLUT1, GLUT2 or GLUT3 levels according to the tissue—brain, retina or liver—considered [94]. Changes in the expression of the mono-carboxylate transporters MCT1 and MCT4, involved in the transport of lactate and pyruvate, were also observed in the brains of mice lacking PrP^C^, with positive or negative regulations depending on the cerebral region considered [95]. In this respect, it is interesting to note that PrP^C^ was found to control the uptake of lactate by astrocytes through the interaction with Na+-K+ ATPase, the driving force for MCT1 activity [96]. Other points of interest include the interaction between PrP^C^ and the lactate dehydrogenase isoforms LDH-A and LDH-B, which were uncovered through systematic proteomic assays for PrP^C^ partners [97,98,99]. PrP^C^ was later shown to potentiate the activity of LDH in the hypoxic brain, which may contribute to the protective role for PrP^C^ against stress [100]. Other PrP^C^ interactors involved in glycolysis include aldolase C (ALDOC) [101] and as well as aldolase A (ALDOA) [99], both catalysing the conversion of fructose-1,6-bisphosphate into glyceraldehyde 3-phosphate (G3P), alpha- and gamma- enolase (ENO1 and ENO2), two isoenzymes that converts 2-phosphoglycerate into phosphoenolpyruvate [97,99], the well-known glyceraldehyde-3-phosphate dehydrogenase (GAPDH) [97,98,99], the triose-phosphate isomerase TPI [98,99] as well as the pyruvate kinases PKM1/PKM2 that catalyse the de-phosphorylation of phosphoenolpyruvate into pyruvate [97,102]. PrP^C^ was further shown to interact with both cytoplasmic malate dehydrogenase [97], which converts oxaloacetate into malate, itself being imported in the mitochondrial matrix, and mitochondrial malate dehydrogenase [99], which catalyses the oxidation of malate into oxaloacetate within the Krebs cycle. It is of note that these overall interactions have been recapitulated in several studies, suggesting functional implications. However, apart from LDH, how PrP^C^ may influence the activity of these diverse enzymes remains to be explored.

On the other hand, several studies have brought to light links between PrP^C^ and mitochondria. First, beyond its main location at the cell surface, PrP^C^ was also found to locate in the mitochondria of healthy mice [103]. PrP^C^ was further shown to co-localise with COX4 [103], one of the nuclear-encoded subunits of complex IV of the respiratory chain. In addition, PrP^C^-deficient mice were reported to have reduced numbers of mitochondria, which have a larger morphology, and an enhanced maximal respiratory capacity, presumably to compensate for low mitochondrial numbers [104,105]. A proteomic study comparing WT and PrP^C^-deficient neurons also exemplified reduced levels of the mitochondrial proteins COX2 in the absence of PrP^C^ [106]. Finally, coming back to PrP^C^ partners, two independent studies have identified citrate synthase (CS) as a PrP^C^ interactor [97,98]. Again, whether PrP^C^ modulates the activity of CS has yet to be investigated. 

As a whole, despite the scarcity of data relating to PrP^C^ and cancer cell metabolism, the findings obtained with respect to PrP^C^ physiological role and, more importantly, to PrP^C^ binding partners offer many avenues for future investigation.

## 8. Evading Immune Destruction (Emerging Hallmark) 

The second emerging hallmark emphasised by Hanahan and Weinberg refers to the ability of cancer cells to thwart the immune system [14]. A first hint at a link between PrP^C^ and immune-evasion can be inferred from the enrichment of *PRNP* gene expression in the mesenchymal subtype of colorectal cancer [28], itself associated with an immune-suppressive signature [107]. Secondly, a more general role for PrP^C^ in immunological quiescence has been proposed, based on its pattern of expression in immune privilege organs as well as its cytoprotective and immune-regulatory function [108]. Mechanistically, PrP^C^ may, by itself, induce cell signalling events sustaining immunomodulation, and thereby temper inflammation [8,108]. On the other hand, tumour associated PrP^C^ controls the levels of several effectors known to promote immune evasion. One such effector is TGFβ, whose soluble levels are regulated by PrP^C^ through a yet-to-be-uncovered mechanism [28]. It is indeed now well-acknowledged that TGFβ inhibits anti-tumour immunity through multiple mechanisms (see [109] for review). Another major player in immune tolerance is IDO (indoleamine 2,2 dioxygenase), an enzyme of the kynurenine pathway [110], which we recently identified as a molecular target downstream from PrP^C^ signalling in colorectal cancer [26]. Thus, it appears that PrP^C^ from cancer cells orchestrates diverse pathways that altogether tone down the anti-tumour immune response by favouring an immune-suppressive contexture. 

## 9. Genome Instability and Mutation 

Genomic alterations represent a key characteristic enabling cancer cells to acquire their diverse hallmarks [14]. The question as to whether PrP^C^ may be linked to this enabling characteristic is twofold. First, we may ask whether genome instability and mutation foster the expression of PrP^C^. A second question is whether the expression of PrP^C^ may afford protection against DNA damage. A major observation regarding the first point is the identification of cell surface PrP^C^ as a marker of aneuploidy in a pan-cancer screening study [111]. The authors further reported an increase in PrP^C^ levels in parental or aneuploid HCT116 colorectal cancer cells upon serum-deprivation, which they linked to oxidative stress, and showed that PrP^C^ is protective against serum-deprivation-induced necrotic death [111]. On this basis, the authors proposed that the upregulation of PrP^C^ in aneuploid cells is a consequence of the oxidative stress associated with this genomic alteration. This notion actually fully fits in with the physiological role described for PrP^C^ in the protection against oxidative stress (reviewed in [6]). According to Qin et al. the induction of *PRNP* gene expression in response to oxidative stress involves the Ataxia Telangiectasia Mutated (ATM) kinase [112]. More recently, ATM was further shown to mediate the upregulation of *PRNP* transcription in response to irradiation [67]. In line with the Domingues study [111], PrP^C^ expression was found to confer a protective role against irradiation [67]. This radioprotective action fits in with the protective role against other types of genotoxic stresses [69], as mentioned above. 

Thus, since PrP^C^ is induced in response to DNA injury and supports protection against DNA damage, we may propose that PrP^C^ takes part in the balance between DNA damage and repair, a trade-off for cancer cell survival and growth [113].

## 10. Tumour-Promoting Inflammation

Chronic inflammation is a well-acknowledged driver of cancer development [114]. Several pieces of evidence indicate that inflammatory conditions may promote an increase in PrP^C^ expression. In the context of cancer, the upregulation of PrP^C^ in Merlin-deficient tumours was found to depend on the NFκB transcription factor [24], which embodies a major link between inflammation and cancer (reviewed in [115]). Whether NFκB positively regulates the expression of PrP^C^ in other types of tumours obviously deserves further investigation. Conversely, PrP^C^ was shown to activate NFκB-dependent transcription in breast cancer cells [82], or to be necessary for the TNFα-dependent activation of NFκB in melanoma M2 cells [116], thereby delineating a bidirectional link between PrP^C^ and NFκB. An alternative yet still hypothetical mechanism leading to an upregulation of PrP^C^ would be via ILK, which we showed to be both a downstream target and an upstream regulator of PrP^C^ [26] and has been reported to be induced in colonic cells in response to inflammation [117]. Another observation worth noting is the induction of PrP^C^ in the mucosa of patients with Helicobacter Pylori gastritis [118], a condition well-known to predispose patients to gastric cancer. 

Paradoxically, *Prnp*-deficient mice were reported to be more sensitive than wild-type mice to dextran sulfate-induced colitis [119]. This is, however, reminiscent of the observations obtained in mice deficient for Yap [120], itself a downstream target of PrP^C^ [28]. In the case of Yap, results were interpreted as Yap being necessary for tissue repair after injury, a function whose over-activation supports cancer progression [121]. Likewise, we may surmise that PrP^C^ is mandatory for tissue regeneration, in accordance with its function in stem cell self-renewal [11], and that its upregulation under inflammatory conditions contributes to cancer development. 

Altogether, the interplay between PrP^C^ and tumour-promoting inflammation is currently supported by few studies and could be worthy of further exploration.

## 11. Conclusions/Future Prospects

In summary, we have enlightened the involvement of PrP^C^ in cancer biology from the standpoint of the hallmarks of cancer (Figure 5). 

While the participation of PrP^C^ to some of those hallmarks—most notably proliferation, survival, invasion and metastasis—is substantiated by multiple cell-based, pre-clinical or clinical studies across cancer types, its links to other hallmarks, such as reprogramming of energy metabolism or evading immune destruction, are only beginning to be explored or even merely suggested from observations outside the field of cancer. These underappreciated roles of PrP^C^ in cancer-related processes represent important areas for future research. Regarding the binding partners involved in the contribution of PrP^C^ to each hallmark (Table 1), data remain scarce at present and are likely to expand in the near future. 

Although some of the signalling pathways through which PrP^C^ operates have been elucidated (Table 2), the picture is far from complete and also requires further investigation. 

Casting further light on these specific points will undoubtedly help reach an integrated view of the multifaceted contribution of PrP^C^ to cancer initiation, promotion and progression. 

## Figures and Tables

**Figure 1 cancers-13-05032-f001:**
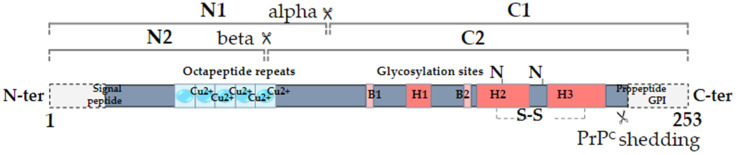
Secondary structure of human PrP^C^. The alpha-cleavage occurs at position 110/111 and generates the N1 and C1 fragments. The beta-cleavage occurs in the vicinity of octapeptide repeats, which bind copper ions, and generates the N2 and C2 fragments. PrP^C^ can also be shed from the plasma membrane through the action of ADAM10. The alpha helices (H) and beta sheets (Β) are indicated (adapted from [7]).

**Figure 2 cancers-13-05032-f002:**
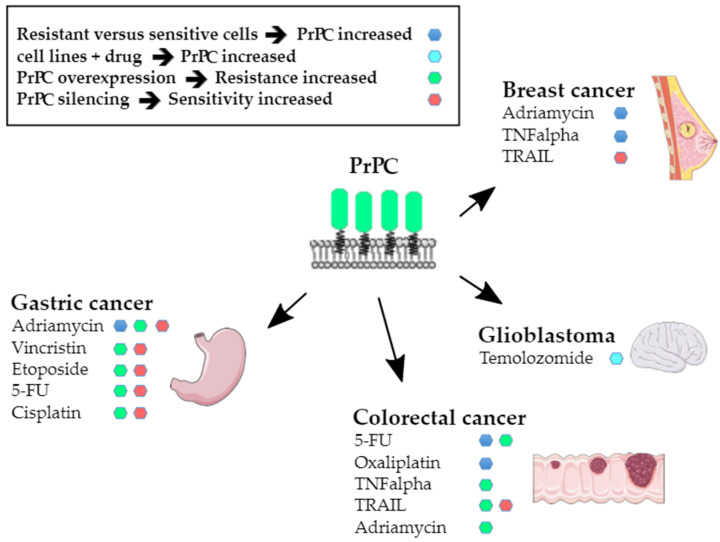
Summary of the data supporting a link between PrP^C^ expression and resistance to cytotoxic agents.

**Figure 3 cancers-13-05032-f003:**
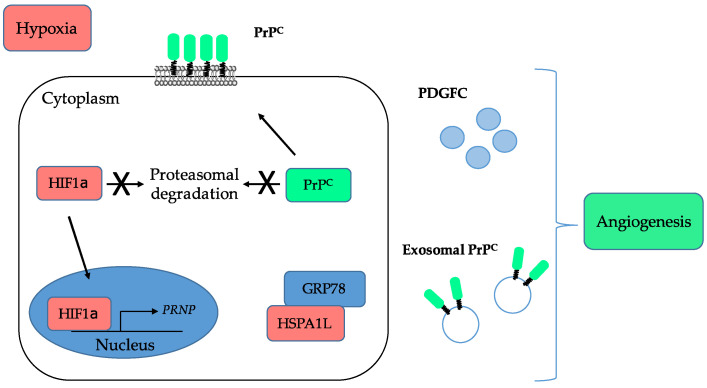
Summary of the interplay between PrP^C^ and hypoxia and of the contribution of PrP^C^ to angiogenesis. PrP^C^ is induced under ischemic conditions and prevents the degradation of HIF1α by the proteasome. On the other hand, under hypoxia, HIF1α promotes the transcription of the *PRNP* gene, favours the sequestration of the GRP78 E3-ligase, leading to enhanced PrP^C^ protein stability, and is associated with enhanced release of exosomal PrP^C^. PrP^C^ may further sustain angiogenesis by controlling the levels of PDGFC.

**Figure 4 cancers-13-05032-f004:**
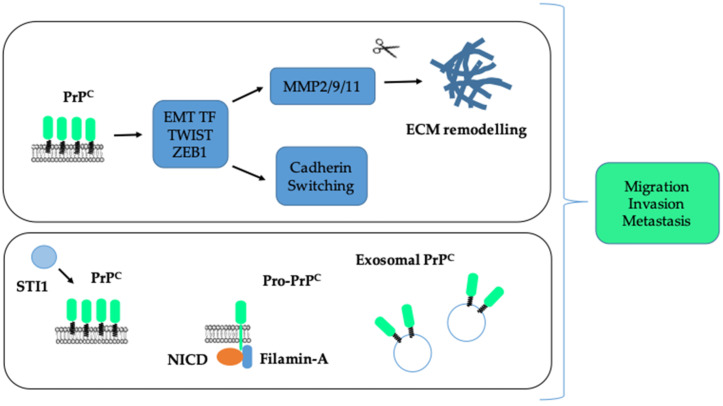
Elevated PrP^C^ is associated with increased invasion and migration of cancer cells and metastasis. PrP^C^ promotes an upregulation of EMT transcription factors (EMT TF), a switch from E-cadherin to N-cadherin expression as well as an induction of matrix metalloproteases, themselves fostering the remodelling of the extracellular matrix (ECM) (top panel). This action may be fostered by the interaction between PrP^C^ and its ligand STI1. It is also supported by pro-PrP^C^, through its interaction with Filamin-A and the active cleaved fragment of Notch, NICD (Notch Intra Cellular Domain) or by exosomal PrP^C^ (bottom panel).

**Figure 5 cancers-13-05032-f005:**
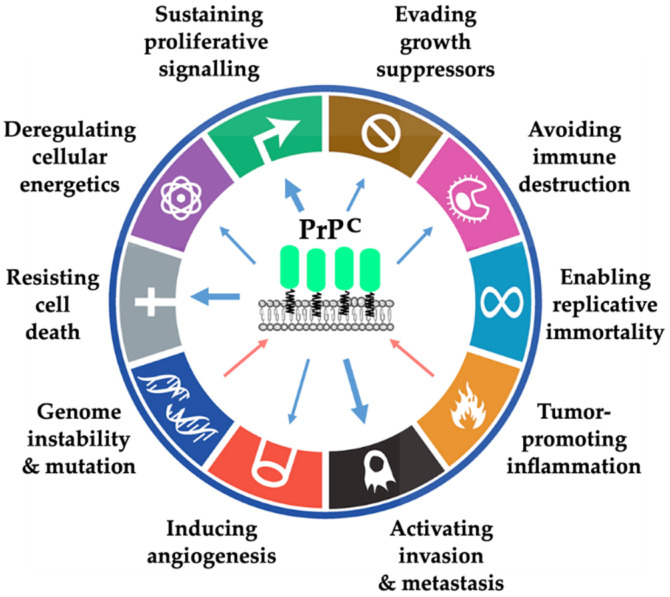
Summary of the data linking PrP^C^ with the different hallmarks of cancer. Blue arrows indicate a contribution of PrP^C^ to the hallmarks and red arrows indicate a regulation of PrP^C^ as a consequence of the hallmark. The thickness of the arrow is related to the amount of evidence supporting the link. The hallmarks of cancer have been adapted from Hanahan and Weinberg [14].

**Table 1 cancers-13-05032-t001:** Summary of the binding partners involved in the PrP^C^ contribution to a given hallmark of cancer, in relation with the cancer type.

Hallmark	Partner	Cancer Type	Reference
Sustaining proliferative signalling	NOTCH1	Pancreatic	[18]
Sustaining proliferative signalling	STI1	Glioblastoma	[21]
Resisting cell death	37LRP	Gastric	[62]
Activating invasion and metastasis	STI1	Colorectal	[87]
Activating invasion and metastasis	Filamin-A	Melanoma	[88]

**Table 2 cancers-13-05032-t002:** Summary of the signalling pathways through which PrP^C^ contributes to a given hallmark of cancer, in relation with the cancer type.

Hallmark	Partner	Cancer Type	Reference
Sustaining proliferative signalling	PI3K/AKT- CyclinD1	Gastric	[15]
Sustaining proliferative signalling	NOTCH1	Pancreatic	[18]
Sustaining proliferative signalling	ILK	Colorectal	[26]
Evading growth suppressors	NF2	Schwannoma	[24]
Evading growth suppressors	TGFβ	Colorectal	[26]
Resisting cell death	MDR1	Gastric	[54]
Resisting cell death	PI3K/AKT	Gastric	[59]
Resisting cell death	BCL2	Gastric	[42,54,58]
Resisting cell death	BCL2	Breast	[51]
Resisting cell death	BCL2	Glioma	[63]
Resisting cell death	Survivin/cIAP-1/XIAP	Colorectal	[25]
Resisting cell death	Par4	Glioblastoma	[23]
Resisting cell death	Soluble PrP^C^	Breast	[66]
Inducing angiogenesis	Hypoxia	Colorectal	[31]
Activating invasion and metastasis	MMPs	Breast	[52,82]
Activating invasion and metastasis	MMPs	Gastric	[83]
Activating invasion and metastasis	NOTCH1	Pancreatic	[18]
Activating invasion and metastasis	JNK	Lung	[84]
Reprogramming of energy metabolism	GLUT1	Colorectal	[29]
Evading immune surveillance	TGFβ	Colorectal	[26]
Evading immune surveillance	IDO	Colorectal	[26]

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
