# Peer review of "The Cellular Prion Protein and the Hallmarks of Cancer"

_cancers, 2021, doi:10.3390/cancers13195032_

Round 1

Reviewer 1 Report

This is a lucid review. I have only minor suggestions/comments:

  1. Fig. 1: Is it possible to use human PrPc instead of that of the mouse as the former seems more relevant for the current review?
  2. In several places, authors use the less commonly adopted term “on another hand” (line 154, 157, 270, and 432). “On the other hand” seems to read more naturally.
  3. Page 11, line 445: “……twofold. First,……..”. After the first point, I was not able to find an obvious second point.
  4. Page 11, line 447: “the Domingues study” comes out of the blue. “the Domingues study [111]” would be better.
  5. Extra spaces: P.5 Line 226, P.6 Line 240, P.6 Line 248, and P.7 Line 290.

Author Response

This is a lucid review. 

We thank the reviewer for his positive comments.

  1. Fig. 1: Is it possible to use human PrPc instead of that of the mouse as the former seems more relevant for the current review?

This is a very relevant point. This has been changed.

  1. In several places, authors use the less commonly adopted term “on another hand” (line 154, 157, 270, and 432). “On the other hand” seems to read more naturally.

This has been changed throughout the text.

  1. Page 11, line 445: “……twofold. First,……..”. After the first point, I was not able to find an obvious second point.

This has been changed by adding “A second question is whether the expression of PrPC may afford protection against DNA damage.”

  1. Page 11, line 447: “the Domingues study” comes out of the blue. “the Domingues study [111]” would be better.

This has been corrected

  1. Extra spaces: P.5 Line 226, P.6 Line 240, P.6 Line 248, and P.7 Line 290.

This has been corrected

Reviewer 2 Report

The authors have reviewed the literature to discuss the possible involvement of prion protein PrPC in relation to the hallmarks of cancer.

Author Response

The authors have reviewed the literature to discuss the possible involvement of prion protein PrPC in relation to the hallmarks of cancer.

We thank the reviewer for his positive comments.

Reviewer 3 Report

The review provides a comprehensive insight to the involvement of PrPc in the various processes that are characteristic of cancer development. The authors report the findings of an impressive number of publications and organize these according to the system of cancer hallmarks. 

The manuscript reports on an actively developing field and supplies important summary of the available knwoledge, highlighting the areas that need further study.

The collected information is ample and detailed, describing various areas of PrPc involvement in cancer progression. However, this diversity makes the understanding of the manuscript somewhat difficult and the reading a tedious process. My main suggestions refer to the addition of summarizing and explanatory tables and/or figures that would facilitate the delivery of the most important messages.

  1. Each sub-chapter describing a hallmark could be supplemented with a  table that organizes the involvement of PrPc with regards to the molecular pathways and proteins that are affected.
  2. A table or figure could be added to the manuscript that demonstrates direct and indirect protein-protein interactions of PrPc and their relations with the different hallmarks.
  3. A figure with the signaling pathways that are influenced by PrPc and their importance in cancer would also be a good idea.

Specific comments:

  1. Information given in lines 458-461 are redundant, the same information was given in lines 253-255 with more-or-less the same wording. I suggest rephrasing one of these or simply referring to the already provided information.
  2. In lines 500-501 it is mentioned that "other hallmarks are only beginning to be explored or even merely suggested from observations outside the field of cancer" - this notion was not overly apparent in the text preceding this paragraph. I suggest a discussion with more emphasis on this aspect, or an additional figure where processes are indicated as having been proven or just suggested by circumferencial evidence.
  3. Resolution of the figures could be improved, especially for Figure 5. Also, many figures use a colour-scheme where white letters are used on a green background - this is very hard to read, it would be better to use a differnet colour setup.

Author Response

The review provides a comprehensive insight to the involvement of PrPc in the various processes that are characteristic of cancer development. The authors report the findings of an impressive number of publications and organize these according to the system of cancer hallmarks. 

The manuscript reports on an actively developing field and supplies important summary of the available knwoledge, highlighting the areas that need further study.

The collected information is ample and detailed, describing various areas of PrPc involvement in cancer progression. However, this diversity makes the understanding of the manuscript somewhat difficult and the reading a tedious process. My main suggestions refer to the addition of summarizing and explanatory tables and/or figures that would facilitate the delivery of the most important messages.

We thank the reviewer for his positive comments.

  1. Each sub-chapter describing a hallmark could be supplemented with a  table that organizes the involvement of PrPc with regards to the molecular pathways and proteins that are affected.
  2. A table or figure could be added to the manuscript that demonstrates direct and indirect protein-protein interactions of PrPc and their relations with the different hallmarks.
  3. A figure with the signaling pathways that are influenced by PrPc and their importance in cancer would also be a good idea.

As suggested by the reviewer, we have added one table summarizing the binding partners of PrPC reported as involved in its contribution to some of the hallmarks of cancer (Table 1), as well as a second table relating to the signalling pathways relaying the action of PrPC (Table 2). Since these tables refer to all hallmarks, they have been inserted at the end of the review.

Specific comments:

  1. Information given in lines 458-461 are redundant, the same information was given in lines 253-255 with more-or-less the same wording. I suggest rephrasing one of these or simply referring to the already provided information.

This has been corrected.

  1. In lines 500-501 it is mentioned that "other hallmarks are only beginning to be explored or even merely suggested from observations outside the field of cancer" - this notion was not overly apparent in the text preceding this paragraph. I suggest a discussion with more emphasis on this aspect, or an additional figure where processes are indicated as having been proven or just suggested by circumferencial evidence.

This is in fact also explained on Figure 5 with corresponding arrows: Blue arrows indicate a contribution of PrPC to the hallmarks and red arrows indicate a regulation of PrPC as a consequence of the hallmark. The thickness of the arrow is related to the amount of evidence supporting the link.

  1. Resolution of the figures could be improved, especially for Figure 5. Also, many figures use a colour-scheme where white letters are used on a green background - this is very hard to read, it would be better to use a differnet colour setup.

We totally agree on this point; all figures have been changed to improve quality.